# Dose-Intense Cisplatin-Based Neoadjuvant Chemotherapy Increases Survival in Advanced Cervical Cancer: An Up-to-Date Meta-Analysis

**DOI:** 10.3390/cancers14030842

**Published:** 2022-02-08

**Authors:** Van Tai Nguyen, Sabine Winterman, Margot Playe, Amélie Benbara, Laurent Zelek, Frédéric Pamoukdjian, Guilhem Bousquet

**Affiliations:** 1National Cancer Hospital, Department of Medical Oncology 1, Hanoi 10000, Vietnam; van-tai.nguyen@inserm.fr; 2Assistance Publique Hôpitaux de Paris, Hôpital Avicenne, Service d’Oncologie Médicale, 93000 Bobigny, France; sabine.winterman@aphp.fr (S.W.); laurent.zelek@aphp.fr (L.Z.); 3Institut National de la Santé et de la Recherche Médicale (INSERM), UMR_S942, Université de Paris-Université Sorbonne Paris Nord, 93000 Bobigny, France; frederic.pamoukdjian@aphp.fr; 4Assistance Publique Hôpitaux de Paris, Hôpital Avicenne, Service de Médecine Nucléaire, 93000 Bobigny, France; margot.playe@aphp.fr; 5Assistance Publique Hôpitaux de Paris, Hôpital Jean Verdier, Service de Gynécologie—Obstétrique, 93140 Bondy, France; amelie.benbara@aphp.fr; 6Université Sorbonne Paris Nord, 9 Avenue Jean Baptiste Clément, 93439 Villetaneuse, France; 7Assistance Publique Hôpitaux de Paris, Hôpital Avicenne, Service de Médecine Gériatrique, 93000 Bobigny, France

**Keywords:** locally advanced cervical cancer, neoadjuvant therapy, dose-intense cisplatin-based

## Abstract

**Simple Summary:**

Although neoadjuvant chemotherapy has become standard care for many locally advanced cancer sites, the benefit of neoadjuvant chemotherapy remains unclear for the treatment of locally advanced cervical cancer. With this meta-analysis, we set out to demonstrate the benefit of using dose-intense cisplatin-based neoadjuvant chemotherapy in terms of overall survival and progression-free survival. Dose-intense cisplatin-based neoadjuvant chemotherapy followed by local therapy was significantly associated with a survival benefit in the treatment of locally advanced cervical carcinoma. Even though radiotherapy combined with weekly cisplatin-based chemotherapy remains standard of care for the treatment of locally advanced cervical cancer, our meta-analysis makes it possible to consider the use of dose-intense cisplatin-based neoadjuvant chemotherapy when local treatment is suboptimal and opens perspectives for designing new clinical trials in this setting.

**Abstract:**

Purpose: We set out to demonstrate the benefit of using dose-intense cisplatin-based neoadjuvant chemotherapy in terms of overall survival and progression-free survival. Methods: We searched through MEDLINE and Cochrane Library databases up to May 2021 to identify randomized clinical trials comparing the benefit of using cisplatin-based neoadjuvant chemotherapy followed by local treatment with local treatment alone for the treatment of locally advanced cervical cancer. The PRISMA statement was applied. Results: Twenty-two randomized clinical trials were retrieved between 1991 and 2019, corresponding to 3632 women with FIGO stages IB2-IVA cervical cancer. More than 50% of the randomized clinical trials were assessed as having a low risk of bias. There was no benefit of neoadjuvant chemotherapy on overall survival, but there was significant heterogeneity across studies (I^2^ = 45%, *p* = 0.01). In contrast, dose-intense cisplatin at over 72.5 mg/m^2^/3 weeks was significantly associated with increased overall survival (RR = 0.87, *p* < 0.05) with no heterogeneity across the pooled studies (I^2^ = 36%, *p* = 0.11). The survival benefit was even greater when cisplatin was administered at a dose over 105 mg/m^2^/3 weeks (RR = 0.79, *p* < 0.05). Conclusion: Even though radiotherapy combined with weekly cisplatin-based chemotherapy remains standard of care for the treatment of locally advanced cervical cancer, our meta-analysis makes it possible to consider the use of dose-intense cisplatin-based neoadjuvant chemotherapy when local treatment is suboptimal and opens perspectives for designing new clinical trials in this setting. Neoadjuvant chemotherapy could be proposed when surgery is local treatment instead of standard chemoradiotherapy for the treatment of locally advanced cervical cancer.

## 1. Introduction

Cervical cancer is the fourth most common cancer among women, with an estimated 604,1270 new cases in 2020 worldwide [1]. It is also the fourth cause of death from cancer among women [1], and 5-year overall survival rates range from 75% to 22% for locally advanced stages IB2 to IVA according to the International Federation of Gynecology and Obstetrics (FIGO) [2]. For locally advanced stages, the standard treatment is whole pelvic radiotherapy combined with cisplatin-based chemotherapy [3,4]. However, barriers to radiotherapy access is a major issue in low-income countries where the incidence and mortality of cervical cancer is higher than in middle- and upper-income countries [5,6,7]. This may be also true in rich countries where daily access to radiotherapy can be difficult in the case of long travel distances [8,9]. In these circumstances, surgery as a local treatment is usually proposed to women. Indeed, for the management of women with invasive cervical cancer, the American Society of Clinical Oncology has recommended that “In basic settings where patients cannot be treated with radiation therapy, extrafascial hysterectomy either alone or after chemotherapy may be an option for women with stage IA1 to IVA cervical cancer” [10].

Neoadjuvant chemotherapy followed by local treatment of primary tumors has become standard care for many locally advanced cancer sites [11,12,13]. However, for cervical cancer, the benefit of neoadjuvant chemotherapy remains controversial, mainly because of the heterogeneous nature of clinical trials, including different clinical stages and various treatment protocols [14]. Cisplatin is the most effective drug for the treatment of cervical cancer, for both locally advanced and metastatic stages [15,16], and resistance to cisplatin is a major cause of relapse and mortality. Cisplatin mainly acts through the formation of platinum-DNA adducts, thus inducing double-strand DNA breaks [17,18]. Cisplatin is particularly efficacious in cancers with DNA-repair pathway defects, including ovarian and breast cancers [19,20]. In contrast, for cervical cancer, the activation of DNA-repair pathways with overexpression of ERCC1, RAD51, or PARP1 may lead to cisplatin resistance [21,22,23]. In the neoadjuvant settings, on patient pretreated biopsies, a high level of HSPB1/p as a molecular marker of DNA damage and repair were associated with a lower response to cisplatin [24]. Higher doses of cisplatin using a dose-intense regimen could be a way to overcome resistance to cisplatin. For metastatic disease, cisplatin-based triplet regimens have improved response rates and survival compared to monotherapy or doublet regimen, but the benefit of using a dose-intense cisplatin-based regimen has not yet been demonstrated [16].

In this meta-analysis, we intended to demonstrate the benefit of using dose-intense cisplatin-based neoadjuvant chemotherapy for the treatment of locally advanced cervical cancer and see if neoadjuvant chemotherapy could be more systematically proposed when surgery is the local treatment instead of standard chemoradiotherapy.

## 2. Materials and Methods

### 2.1. Search Strategy and Selection Criteria

The Preferred Reporting Items for Systematic reviews and Meta-Analyses (PRISMA) method was applied for this meta-analysis [25]. For the search strategy, we applied the following method: using an ad hoc algorithm composed of both thesaurus and free-text terms, we searched MEDLINE via PubMed and Cochrane Library for articles published up to May 2021. In addition, the references contained in the articles and relevant reviews identified were also considered to avoid eligible articles being missed. The algorithm was the following: (“Uterine Cervical Neoplasms” [MeSH] OR “Cervix Neoplasms” OR “Cervix Cancer” OR “Cervical Neoplasm”) AND (“Locally Advanced”) AND (“Neoadjuvant Therapy”[MeSH] OR “Neoadjuvant Chemotherapy”). The International Prospective Register of Systematic Review (PROSPERO) study’s Registration Number CRD42021245170 in April 2021.

For study selection, we applied the Population, Intervention, Comparator group, Outcomes, and Study design (PICOS) criteria [25] as detailed in Appendix A. The studies that were included in this meta-analysis met the following inclusion criteria: (i) all studies were randomized clinical trials; (ii) the patients had stages IB2 to IVA cervical cancer; (iii) the patients received or did not receive cisplatin-based neoadjuvant chemotherapy followed by local treatment including chemoradiotherapy or radiotherapy alone or surgery alone; (iv) the study endpoints were overall survival and progression-free survival. Overall survival (OS) was defined as the time interval between the date of randomization and the date of death from any cause or the last follow-up. Progression-free survival (PFS) was defined as the time from randomization until the first confirmation of progression or death from any cause.

Two authors (V.T.N. and G.B.) independently screened the papers retrieved, initially by titles, then by abstracts, and finally by full texts. In the case of discordance, a third author (F.P.) was solicited to make the decision whether or not to retrieve the paper. We identified a total of 22 relevant publications. The quality of the randomized clinical trials was evaluated using the Cochrane Handbook for Systematic Reviews of Interventions (CHSRI) guidelines, and the risk of bias was classified as low, unclear, or high [26].

### 2.2. Statistical Analyses

The data were analyzed using R statistical software (version 4.0.3; R Foundation for Statistical Computing, Vienna, Austria; http://www.r-project.org, accessed on 27 March 2021). Categorical variables were summarized as numbers (percentage), and continuous variables were summarized as means ± standard deviation (SD) or medians ± interquartile range (IQR). On the articles selected, we performed a meta-analysis (with the package “meta”) to assess the prognostic value of neoadjuvant cisplatin-based chemotherapy in locally advanced cervical cancers for overall survival (OS) and progression-free survival (PFS). We assessed the heterogeneity of study results by using the I^2^ indicator and Cochran’s Q test. I^2^ values of 0%, 25%, 50%, and 75% were considered to indicate absence of heterogeneity and low, moderate, and high heterogeneity, respectively. A value of *p* ≤ 0.05 on the Q test indicated significant heterogeneity. Graphically, the pooled results were summarized as risk ratios (RRs) and their 95% confidence interval (95%CI) in a forest plot using fixed or random effect as appropriate. Publication bias was assessed both graphically using a funnel plot and quantitatively with a linear regression test of funnel plot asymmetry (nonsignificant *p* value meaning no publication bias). All tests were two sided, and the threshold for statistical significance was set at a value of *p* < 0.05.

## 3. Results

The literature search and screening processes are detailed in Figure 1. The search algorithm initially identified 426 articles, and 12 additional articles were considered after reading the references or related reviews. After careful screening, 416 articles were excluded, most of them being nonrandomized trials (74) or trials with a nonrelevant outcome (108). The articles excluded at the full-text screening stage are listed in Appendix A. Twenty-two articles were finally retrieved for this meta-analysis.

These 22 eligible randomized studies (14 phase II and 8 phase III clinical trials) included a total of 3632 women with stages IB2 to IVA cervical cancers. The study characteristics are listed in Table 1. Mean/median ages ranged from 39 to 57 years in the neoadjuvant chemotherapy arm, and the median follow-up time ranged from 16 to 108 months. Briefly, the local treatment was chemoradiotherapy in one study, radiotherapy in fourteen studies, and surgery in seven studies. We found that the three studies by Sardi et al. [27,28,29], and the two studies by Tattersall et al. [30,31], were not overlapping and thus did not concern the same patients included in these five clinical trials conducted by two different teams. Concerning patients treated with neoadjuvant chemotherapy, all received a cisplatin-based regimen with a planned total dose of cisplatin ranging from 100 to 300 mg/m^2^ over a period ranging from 28 to 63 days.

Regarding progression-free survival, only eighteen of the studies included reported these data, with 1211 events. There was no significant heterogeneity in the overall analysis (I^2^ = 28%, *p* = 0.13), and cisplatin-based neoadjuvant chemotherapy was significantly associated with increased PFS (RR for fixed effect = 0.9 [0.83–0.98]) (Figure 2). There was no significant publication bias (*p* = 0.79) (Appendix A).

After a median follow-up ranging from 16 to 108 months, 1453 deaths had occurred. Regarding overall survival, because of significant heterogeneity in the overall analysis (I^2^ = 45%, *p* = 0.01) and no significant fixed or random effect (Appendix A), we further pooled studies according to dose-intense cisplatin-based neoadjuvant chemotherapy. Dose-intense chemotherapy was calculated as the ratio of the total dose of cisplatin (mg/m^2^)/chemotherapy duration (weeks) and was expressed in mg/m^2^/3 weeks. Dose-intense cisplatin administration ranged from 50 to 160 mg/m^2^/3 weeks with a median value of 72.5 mg/m^2^/3 weeks. After a sensitivity analysis using the threshold of 72.5, and then 105 mg/m^2^/3 weeks, we classified studies in a binary manner as a function of dose-intense cisplatin use (yes/no) at the chosen thresholds (i.e., < or ≥) to maximize both fixed and random effects and to reduce heterogeneity. Dose-intense cisplatin ≥ 72.5 mg/m^2^/3 weeks was significantly associated with OS (RR for fixed effect = 0.87 [0.76–0.98]). In contrast, a dose-intense cisplatin < 72.5 mg/m^2^/3 weeks was significantly deleterious (RR = 1.15 [1.02–1.28]) (Appendix A). The benefit was even greater at dose-intense cisplatin ≥ 105 mg/m^2^/3 weeks (RR for fixed effect = 0.79 [0.67–0.93]) (Figure 3).

When we restricted the analysis to subgroups of patients whose local treatment was similar (surgery or radiotherapy), dose-intense cisplatin was beneficial in both cases and more marked when local treatment was limited to a surgery (Appendix A).

In contrast, when we pooled studies according to (i) triplet vs. not triplet cisplatin-based neoadjuvant chemotherapy or (ii) duration of chemotherapy (≤ vs. >6 weeks), there was no significant difference in terms of overall survival (Appendix A).

For PFS, we performed the same subgroup analyses. In particular, the benefit of using cisplatin-based neoadjuvant chemotherapy was also significantly increased when cisplatin dose-intensity was higher than 72.5 mg/m^2^/3 weeks (RR for fixed effect = 0.86 [0.78–0.95]). The RR of 0.84 was comparable in terms of PFS for the subgroup of studies with surgery as local treatment and receiving neoadjuvant chemotherapy, despite that it did not reach statistical significance. There was no heterogeneity between the subgroups (Appendix A).

Table 2 shows the RR for fixed effect in the sensitivity analysis. There was no significant publication bias (*p* = 0.73) (Appendix A), and the overall quality of the studies included in this meta-analysis was good according to the Cochrane Collaboration Risk of Bias tool (Appendix A). In all, more than 50% of the randomized clinical trials were at low risk of bias.

Overall, dose-intense cisplatin-based neoadjuvant chemotherapy was significantly associated with a survival benefit in the treatment of locally advanced cervical carcinoma, and the higher the dose, the greater the survival benefit.

## 4. Discussion

For the first time, our meta-analysis has demonstrated the benefit of using dose-intense cisplatin-based neoadjuvant chemotherapy for the treatment of locally advanced cervical cancer. Two meta-analyses had previously been conducted in this setting. In 2020, the second meta-analysis only focused on stages IB2 and II cervical cancers and was thus limited to 5 studies, including 2 case–control studies and only 1275 patients, with a nonsignificant trend for a 40% reduction in the risk of death with neoadjuvant chemotherapy [50]. In 2003, the first meta-analysis reported data for 2074 women without evidencing any benefit of neoadjuvant chemotherapy. However, when the authors analyzed studies using dose-intense cisplatin separately for doses over 25 mg/m^2^/week (i.e., over 75 mg/m^2^/3 weeks), there was a nonsignificant trend towards a survival benefit (HR = 0.91; *p* = 0.2) [14]. In our meta-analysis, using dose-intense cisplatin at a dose over 72.5 mg/m^2^/3 weeks, the RR for overall survival was 0.87 (*p* < 0.05). Using dose-intense cisplatin at doses over 105 mg/m^2^/3 weeks, the benefit was even greater with a RR of 0.79. In contrast, using cisplatin at a dose less than 72.5 mg/m^2^/3 weeks was significantly associated with shorter survival, possibly related to the delay in instating local treatment. This was also true for PFS. Since 2003, 8 additional published trials have been reported, corresponding to 1345 additional patients, enabling the statistical power of our meta-analysis to be increased.

The stringent methodology is a strength of our meta-analysis on published data, with clearly defined PICOS criteria, careful selection of the clinical trials finally retained, absence of heterogeneity across the subgroups compared, and quality control of the 22 studies to ensure the absence of publication bias. A limitation of this meta-analysis could be linked to the local treatments instated after neoadjuvant chemotherapy, since radiotherapy combined with concomitant chemotherapy, the standard care, only concerned one clinical trial. Most local treatments were either surgery alone or radiotherapy alone. However, dose-intense cisplatin-based neoadjuvant chemotherapy remained beneficial when we analyzed these subgroups separately, and there was no heterogeneity between the subgroups of patients receiving or not receiving cisplatin-based dose-intense neoadjuvant chemotherapy, independently from the local treatment performed after neoadjuvant chemotherapy. In particular, two clinical trials included in this meta-analysis raise the concern that local treatment instated after neoadjuvant chemotherapy was not the same between the experimental arm and the control arm: the studies by Benedetti-Panici et al. [40] and Chang et al. [41]. Even after exclusion of these two trials from our meta-analysis, the results were not different with a survival benefit of using dose-intense cisplatin-based neoadjuvant chemotherapy (Appendix A).

In addition, one should keep in mind that radiotherapy is not available in many countries. This is particularly true in low-income countries, accounting for more than 80% of cervical cancers diagnosed worldwide, where the incidence of locally advanced cervical cancer is the highest. In these countries, surgery is a frequent therapeutic alternative to chemoradiotherapy when it is not available [5,6,51], and the American Association Society has recommended that “In basic settings where patients cannot be treated with radiation therapy, extrafascial hysterectomy either alone or after chemotherapy may be an option for women with stage IA1 to IVA cervical cancer” [10]. This is also true in upper- and middle-income countries when daily access to radiotherapy is difficult due to long travel distances [8,9]. In all these cases, cisplatin-based dose-intense neoadjuvant chemotherapy could be beneficial when local treatment is suboptimal, both in terms of PFS and OS. In a previous meta-analysis, Ye et al. demonstrated the benefit of neoadjuvant chemotherapy when local treatment was surgery [50]. One should keep in mind that when surgery is chosen as a local treatment for cervical cancer with high-risk characteristics, adjuvant radiotherapy should be proposed as standard care despite that its access may be often very difficult [10].

One concern may be the clinical and limited toxicities when using high doses of cisplatin. These toxicities, mainly hematological and gastrointestinal toxicities, are usually easily manageable using new-generation antiemetics and granulocyte colony-stimulating factor (G-CSF) [52,53,54].

In metastatic stages, triplet regimens of chemotherapy lead to higher response rates compared to doublet regimens and to monotherapy, but survival data were insufficient to reach any relevant conclusion [16]. In our meta-analysis, we did not find any survival benefit of using a cisplatin-based triplet regimen for neoadjuvant chemotherapy in locally advanced cervical cancer. Further studies are required to assess this point, including the addition of a taxane to cisplatin, since only one clinical trial has evaluated the combination of paclitaxel and carboplatin in the neoadjuvant setting, evidencing no clear benefit [55].

## 5. Conclusions

Even though radiotherapy combined with weekly cisplatin-based chemotherapy remains standard of care for the treatment of locally advanced cervical cancer, our meta-analysis makes it possible to consider the use of dose-intense cisplatin-based neoadjuvant chemotherapy when local treatment is suboptimal and opens perspectives for designing new clinical trials in this setting, particularly for FIGO stages III and IVA. Neoadjuvant chemotherapy could be proposed when surgery is local treatment instead of standard chemoradiation therapy for the treatment of locally advanced cervical cancer, as proposed in the EORTC 55994 trial design [56] (Figure 4).

## Figures and Tables

**Figure 1 cancers-14-00842-f001:**
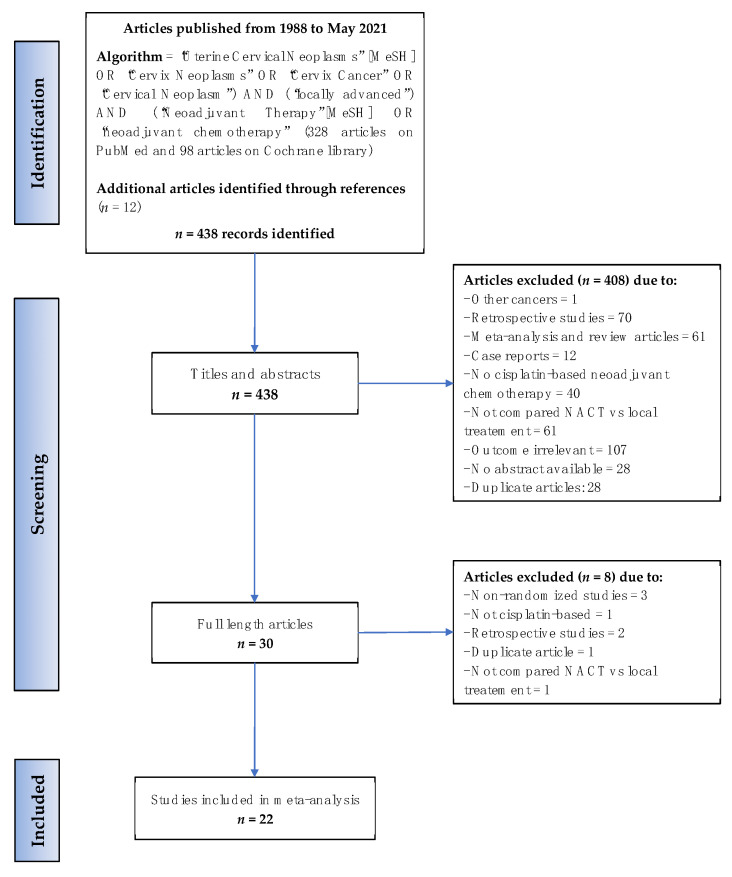
Flow diagram of the search strategy. MeSH, Medical Subject Headings; NACT, neoadjuvant chemotherapy.

**Figure 2 cancers-14-00842-f002:**
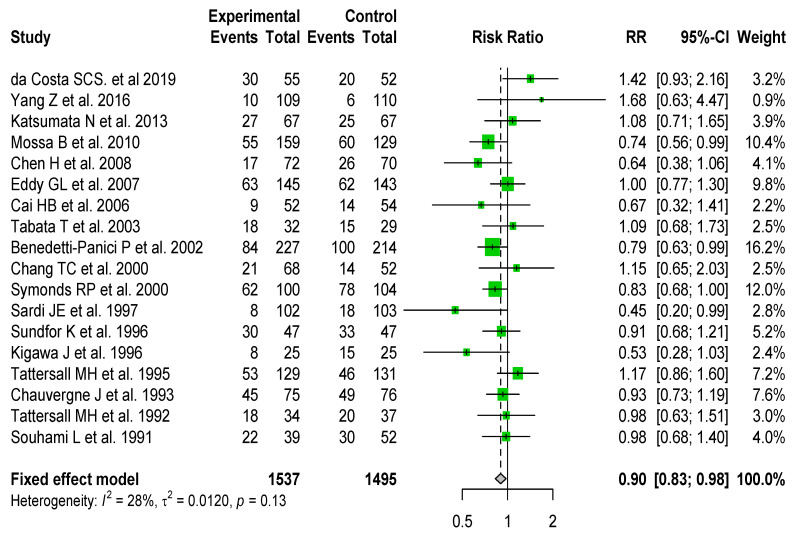
Forest plot of randomized clinical trials assessing predictive value of cisplatin-based neoadjuvant chemotherapy on progression-free survival in locally advanced cervical cancers. Relative risks (RR) are given with 95% confidence intervals. Overall RR = 0.9 (95% CI, 0.83 to 0.98); heterogeneity I^2^ = 0.28, *p* = 0.13.

**Figure 3 cancers-14-00842-f003:**
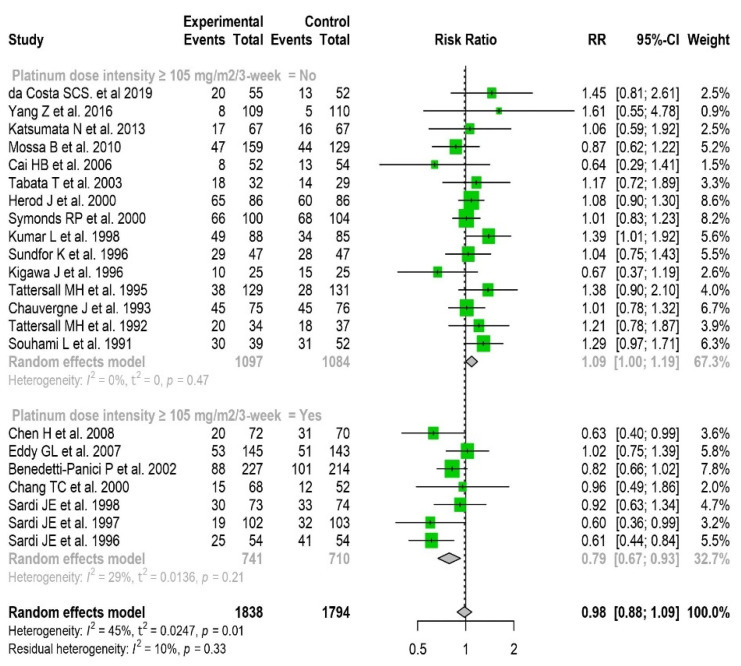
Forest plot of randomized clinical trials assessing the predictive value of neoadjuvant cisplatin-based chemotherapy for overall survival in locally advanced cervical cancer. Relative risks (RR) are given with 95% confidence intervals. Neoadjuvant chemotherapy using dose-intense cisplatin-based chemotherapy ≥ 105 mg/m^2^/3 weeks significantly improves overall survival: RR = 0.79 (95% CI, 0.67 to 0.93); heterogeneity I^2^ = 29%, *p* = 0.21.

**Figure 4 cancers-14-00842-f004:**
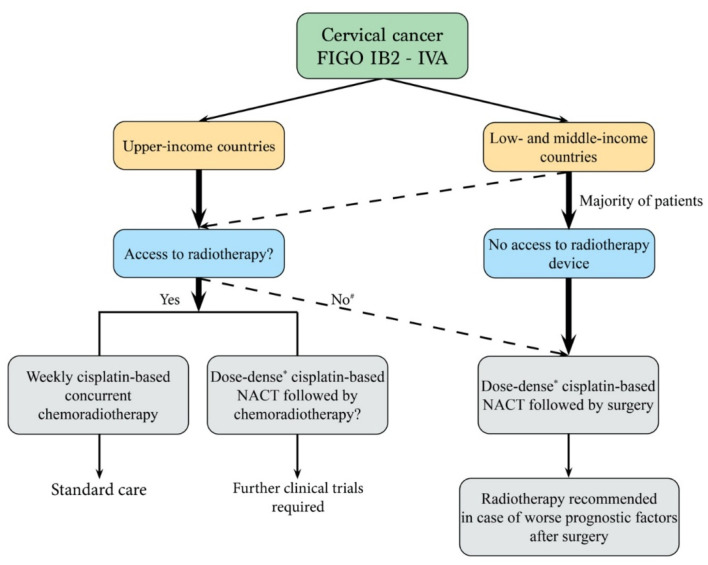
Decision tree for the treatment of locally advanced cervical cancer. NACT, neoadjuvant chemotherapy; ^#^ vulnerable populations, long travel distances; * minimal cisplatin dose of 75 mg/m^2^/3 weeks.

**Table 1 cancers-14-00842-t001:** The main characteristics of the 22 studies included in the meta-analysis.

Country Author Year Trial	Median Follow-up (Months)	Comparison	Experimental * Arm		Control Arm **		Figostage	NACT Regimen	Planned Duration of NACT	Cisplatin (mg/m^2^/3 Weeks)	Radiotherapy
			n	Age ^#^	n	Age ^#^					
Brazil da Costa [32] 2019 Phase II	31.7	NACT+CRT vs. CRT	55	48 (22–69)	52	45 (20–67)	IIB-IVA ^a^	CCDP 50 mg/m^2^ Gem 1000 mg/m^2^, Day 1, 8	Every 21 days for 3 cycles	50	Weekly CCDP 40 mg/m^2^ with RT 45–50.4 Gy over 6 weeks followed by BT 28–30 Gy
China Yang [33] 2016 Phase III	32	NACT+surgery vs. surgery	109	47 (23–66)	110	48 (26–68)	IB2-II ^a^	CCDP 70 mg/m^2^ IRI 180 mg/m^2^ OR CCDP 70 Pacl 175 mg/m^2^	Every 21 days for 1–2 cycles	70	Postoperative RT of 48–50 Gy if risk factors.
Japan Katsumata [34] 2013 Phase III	49	NACT+surgery vs. surgery	67	47 (28–70)	67	46 (22–67)	IB2-IIB ^b^	CCDP 70 mg/m^2^ BLM 35 mg/m^2^ VCR 0.7 mg/m^2^ MMC 7 mg/m^2^	Every 21 days for 2 cycles	70	Postoperative RT of 45–50.4 Gy if risk factors. BT if surgical margins positive
Italy Mossa [35] 2010 Phase III	84	NACT+surgery/RT vs. surgery/RT	159	48.5 (32–65)	129	48.5 (32–65)	IB-IIIB ^b^	CCDP 50 mg/m^2^ VCR 1 mg/m^2^ BLM 75 mg/m^2^	Every 21 days for 3 cycles	50	50 Gy over 5–6 weeks followed by BT a maximum 30 Gy
China Chen [36] 2008 Phase II	NA	NACT+surgery vs. surgery	72	44 (25–74)	70	44 (25–74)	IB2-IIB ^b^	CCDP 100 mg/m^2^ MMC 20 mg/m^2^ 5FU 120 mg/kg	Every 14 days for 2–3 cycles	150	Postoperative pelvic RT at a dose of 45 Gy if risk factors after surgery
American Eddy [37] 2007 Phase III	62	NACT+surgery vs. surgery	145	NA	143	NA	IB2 ^b^	CCDP 50 mg/m^2^ VCR 1 mg/m^2^	Every 10 days for 3 cycles	105	Postoperative pelvic RT if risk factors after surgery
China Cai [38] 2006 Phase II	62	NACT+surgery vs. surgery	52	45.6	54	44.8	IB ^b^	CCDP 75 mg/m^2^ 5FU 120 mg/kg	Every 21 days for 2 cycles	75	Postoperative RT at a dose of 45 Gy if risk factors after surgery
Italy Tabata [39] 2003 Phase II	NA	NACT+RT vs. RT	32	57 (35–68)	29	59 (44–70)	IIIB-IVA ^b^	CCDP 70 mg/m^2^ BLM 5 mg/m^2^ VCR 0.7 mg/m^2^ MMC 7 mg/m^2^	Every 28 days for 3 cycles	52.5	50 Gy in 25 F followed by BT
Italy Benedetti-Panici [40] 2002 Phase III	40	NACT+surgery vs. RT	227	49 (25–70)	214	52 (28–69)	IB2-III ^b^	CCDP 160 mg/m^2^ BLM 15 mg/m^2^, Day 1, 8	Every 21 days for 2 cycles	160 ^§^	Median total dose of 70 Gy delivered to point A over 62 days
China Chang [41] 2000 Phase II	39	NACT+surgery vs. RT	68	46 (33–69)	52	47 (32–70)	IB2-IIA ^b^	CCDP 50 mg/m^2^ VCR 1 mg/m^2^ BLM 75 mg/m^2^	Every 10 days for 3 cycles	105	50–54 Gy followed by BT, or 70 Gy without BT
England Herod [42] 2000 Phase III	108	NACT+RT vs. RT	86	47 (24–74)	86	46 (27–73)	IB-IVA ^b^	CCDP 50 mg/m^2^ BLM 30 mg IFOS 5 g/m^2^ Mesna 6 g/m^2^	Every 21 days for 2–3 cycles	50	According to institutional policy
England Symonds [43] 2000 Phase III	65	NACT+RT vs. RT	100	49 (25–69)	104	48 (24–70)	IIB-IVA ^b^	CCDP 50 mg/m^2^ MTX 100 mg/m^2^	Every 14 days for 3 cycles	75	40–45 Gy in 20 F over 28 days followed by BT 24–33.75 Gy
Argentina Sardi [29] 1998 Phase II	84	NACT+RT vs. RT	73	42.9	74	41.5	IIB ^c^	CCDP 50 mg/m^2^ VCR 1 mg/m^2^ BLM 75 mg/m^2^	Every 10 days for 3 cycles	105	50–60 Gy in 28–30 F over 45–50 days, followed by BT 25–35 Gy
India Kumar [44] 1998 Phase II	NA	NACT+RT vs. RT	88	45 (30–65)	85	45.5 (21–65)	IIB-IVA ^c^	CCDP 50 mg/m^2^ IFOS 5 g/m^2^ BLM 15 mg/m^2^ Mesna 3 g/m^2^	Every 21 days for 2 cycles	50	40 Gy in 22 F + 10 Gy in 5 F over 35 days followed by BT 30 Gy
Argentina Sardi [28] 1997 Phase II	67	NACT+ surgery ± RT vs. Surgery±RT	102	39 (23–68)	103	41 (24–69)	IB ^c^	CCDP 50 mg/m^2^ VCR 1 mg/m^2^ BLM 75 mg/m^2^	Every 10 days for 3 cycles	105	50–60 Gy over 45–50 days followed by BT 25–35 Gy
Argentina Sardi [27] 1996 Phase II	28	NACT+RT vs. RT	54	48.2	54	49.6	IIIB ^c^	CCDP 50 mg/m^2^ VCR 1 mg/m^2^ BLM 75 mg/m^2^	Every 10 days for 3 cycles	105	50–60 Gy over 45–50 days followed by BT 25–35 Gy
Norway Sundfor [45] 1996 Phase II	46	NACT+RT vs. RT	47	52.7 (25–70)	47	52.2 (26–70)	IIIB-IVA ^c^	CCDP 100 mg/m^2^ 5FU 5000 mg/m^2^	Every 21 days for 3 cycles	100	64.8 Gy in 36 F over 50 days
Japan Kigawa [46] 1996 Phase II	42	NACT ^&^±surgery ±RT vs. RT	25	55.6 (41–67)	25	60.2 (43–69)	IIB-IIIB ^c^	CCDP 50 mg/m^2^ BLM 30 mg/m^2^	Every 21 days for 2–3 cycles	50	50 Gy in 25 F over 35 days followed by BT 24–38 Gy
Australia Tattersall [30] 1995 Phase II	16	NACT+RT vs. RT	129	47 (26–75)	131	52 (27–78)	IIB-IVA ^c^	CCDP 60 mg/m^2^ EPI 110 mg/m^2^	Every 21 days for 2–3 cycles	60	40–55 Gy over 28–35 days followed by BT 30–35 Gy
France Chauvergne [47] 1993 Phase III	84	NACT+RT vs. RT	75	54.3	76	54	IIB-IIIB ^c^	CCDP 80 mg/m^2^ CLB 20 mg/m^2^ VCR 0.7 mg/m^2^ MTX 30 mg/m^2^	Every 21 days for 2–4 cycles	80	45 Gy followed by BT
Australia Tattersall 1992 [31] Phase II	37	NACT+RT vs. RT	34	54 (33–70)	37	56 (23–70)	IIB-IVA ^c^	CCDP 50 mg/m^2^ VBL 4 mg/m^2^ BLM 45 mg/m^2^	Every 21 days for 3 cycles	50	40–55 Gy in 20–25 F over 28–35 days
Brazil Souhami [48] 1991 Phase II	44	NACT+RT vs. RT	39	50 (24–69)	52	49 (26–69)	IIIB ^c^	CCDP 50 mg/m^2^ VCR 1 mg/m^2^ BLM 120U MMC 10 mg/m^2^	Every 21 days for 3 cycles	50	50 Gy in 25 F over 35 days followed by BT 40 Gy

Abbreviations: NACT, neoadjuvant chemotherapy; RT, radiotherapy; CRT, chemoradiotherapy; FIGO, International Federation of Gynecology and Obstetrics; CDDP, cisplatin; Gem, gemcitabine; IRI, irinotecan; Pacl, paclitaxel; VCR, vincristine; BLM, bleomycin; VBL, vinblastine; IFOS, ifosfamide; 5-FU, 5-fluorouracil; MMC, mitomycin; CLB, chlorambucil; EPI, epirubicin; F, fractions; BT, brachytherapy. NA, not available. ^#^ Age values are presented as median or mean age with minimum and maximum values. * Experimental arm: neoadjuvant chemotherapy arm. ** Control arm: local treatment arm. ^&^ Neoadjuvant intra-arterial infusion chemotherapy. ^§^ The trial used 4 different cisplatin-based regimens with a median cisplatin total dose of 300 mg/m^2^ over median 39 days. ^a^ FIGO staging 2009. ^b^ FIGO staging 1994. ^c^ FIGO staging 1988. None of the studies included in the meta-analysis have used the last 2018 FIGO staging [49], and the previous classifications were versions of 1988, 1994, and 2009 FIGO staging.

**Table 2 cancers-14-00842-t002:** Sensitivity analysis of randomized clinical trials assessing the predictive value of cisplatin-based neoadjuvant chemotherapy for overall survival.

	RR for Fixed Effect	[95%CI]
Overall analysis	0.97	[0.90–1.05]
Dose-intense cisplatin ≥ 72.5 mg/m^2^/3 weeks	0.87	[0.76–0.98]
Dose-intense cisplatin ≥ 105 mg/m^2^/3 weeks	0.79	[0.67–0.93]
Triplet cisplatin-based chemotherapy (yes)	0.97	[0.83–1.13]
Chemotherapy duration (≤6 weeks)	0.91	[0.80; 1.04]

RR, relative risk; CI, confidence interval.

## Data Availability

All the data presented in this study are identified within the text, tables, and figures.

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
