# Peer review of "Dose-Intense Cisplatin-Based Neoadjuvant Chemotherapy Increases Survival in Advanced Cervical Cancer: An Up-to-Date Meta-Analysis"

_cancers, 2022, doi:10.3390/cancers14030842_

Round 1

Reviewer 1 Report

General statement:

Authors aptly underwent a robust and clean methodology for the current meta-analysis and built over older similar studies in order to reach their outcomes that were more reliable due to higher number of patients, with minimum bias and heterogeneity. Results were re-enforced by appropriate sub-group analyses in supplementary material. Restricting neoadjuvant chemotherapy to cisplatin only containing regimens excluded a very imminent recent trial (Gupta, 2018) with over 600 patients that were randomized to receive standard of care (definitive chemoradiotherapy) vs neoadjuvant chemotherapy (carboplatin/paclitaxel) followed by surgery after assessment of response. That may have altered the results if included. With only a single study in the meta-analysis (da Costa, 2019) comparing standard of care (definitive chemoradiotherapy) with or without neoadjuvant chemotherapy, it is not appropriate to suggest that neoadjuvant chemotherapy followed by local therapy can an alternative to standard of care if radiotherapy is not adequate. Over and above, if neoadjuvant chemotherapy followed by surgery is used as a suggested option, still adjuvant radiotherapy would be needed for high-risk features as stated in all of the trials included in this submission. Last but not least, the use of term ‘dose-dense cisplatin’ seems misleading to audience because it usually refers to standard dose administered in a shorter time (eg. 75mg/m2 cisplatin every 2 weeks instead of every 3 weeks). If we apply this definition, we will only have 7 out of the included 22 studies to be stated as receiving a dose-dense chemotherapy protocol. Cisplatin doses >105mg/m2 every 3 weeks are not included in current guidelines and were mainly used in the trials with dose-dense cisplatin except for Benedetti-Panici, 2002. All of these concerns need to be addressed before resubmission in addition to the following. 

Simple summary and Abstract:

  • Line 21: kindly replace ‘most’ by ‘many’ and do the same in introduction
  • Line 24: ‘dose-dense’ should be exchanged for ‘dose-intense’ or ‘high-dose’ as the term in this setting is misleading (kindly see above and maintain throughout the submission)
  • Line 25: please add ‘neoadjuvant chemotherapy followed by local therapy’
  • Line 29: we need to fine tune this conclusion as discussed above

Introduction:

  • Lines 53, 54: reference 1 would be better replaced by [Hyuna Sung, 2021; PMID: 33538338] to reflect 2020 update
  • Line 59: replace ‘most’ with ‘many’
  • Line 59: please remove reference 8 as neoadjuvant chemotherapy is not a well-established option for head and neck cancers, authors may add other reference for rectal cancer if they would like

Methods:

Methodology was adequate and clearly explained. Criteria for inclusion/exclusion were also logic and consistent.

  • Line 84: as all of the included studies were published before the 2018 update for the FIGO staging, authors need to confirm that they are using the previous FIGO staging for cervical cancer (2009) and a reference is needed
  • Lines 89-91: the endpoint progression free survival can be better replaced by disease free survival or relapse free survival and this fits the stated definition

Results:

  • Table 1 (FIGO staging 2009 needs to be clarified)
  • Lines 147-148: you can either remove this endpoint and restrict analysis for OS or provide similar sub-analyses for this endpoint as done for OS (Subgroup analysis for surgery/RT and low/high dose cisplatin)

Discussion:

The authors compared their work to the previous 2 similar meta-analyses in a clear interesting way that clarified major differences. Latter on, they discussed possible confounders and limitations.

  • Lines 224-232: Please review the whole paragraph: Addressing problems with reaching radiotherapy in countries with limited resources as a surrogate to advocate neoadjuvant chemotherapy followed by surgery seems not accurate. Classically following recommendations of authors is not a guarantee that no risk factors will be detected after surgery that would mandate adjuvant radiotherapy and this needs clarification. Even when restricting inclusion to FIGO-2009 stages IB2-IIB, Gupta et al, 2018; demonstrated that 21.5% crossed to CRT before surgery and another 20% had risk factors after surgery and received adjuvant RT.
  • Line 239: reference 44 is reference 10 by error (kindly correct and reorder)

Conclusions:

 Kindly review the general statement and edit.

Reviewer 2 Report

The current review focuses on chemotherapy in cervical cancer and this meta analysis is a state-of-art that can be ideal for scientists working in field of cancer therapy and chemotherapy. I suggest publication of current manuscript upon minor revision. The references are not up to date and more newly published articles from 2020 and 2021 should be added to improve quality and visibility of article. The conclusion should be extended and elaborated. An abbreviation section should be added at the end of manuscript before references. The first paragraph of introduction is about cervical cancer. It is too general and more information should be added and specific information are required. The part related to cisplatin in introduction is not good and authors should also discuss mechanism of action of cisplatin and chance of resistance (Doi, 10.1002/ptr.7305; Doi, 10.3390/molecules26082382). It is possible to add a schematic figure to improve quality of article? At the end of introduction, authors have provided aim of review. It is concise and should be extended. 

Round 2

Reviewer 1 Report

I would like to thank the authors for the great efforts they exerted to improve their submission and address reviewers’ concerns. This has added much to the quality of the presented data and supported the conclusions/recommendations. Nevertheless, I still have some concerns regarding this revised submission.

1] I am more conservative for the use of dose-dense to describe doses of cisplatin>72.5 mg/m2 and I am not convinced with the explanation offered by authors in their response. Again, I prefer using dose intense regimen guided by the NACCCMA 2003 meta-analysis.

A-According to Kumar, 2015 (PMID: 25455846)

[[ ‘’‘The delivery of a given amount of drug at shorter time intervals than is standard, while maintaining the per-cycle dose and overall dose, has been referred to as dose-dense therapy [21, 22] . Dose intensity refers to the dose (per unit weight or body surface area) delivered per unit time (e.g. mg/m 2 /week). Therefore, dose intensity can be modified by either increasing the dose per cycle, or by reducing the time between treatments (increasing dose density).’’

  1. National Cancer Institute, US National Institute of Health. Dictionary of Cancer Terms.
  2. Simon R., Norton L.: The Norton-Simon hypothesis: designing more effective and less toxic chemotherapeutic regimens. Nat Clin Pract Oncol 2006; 3: pp. 406-407. ]]

B- In a study: Robova H, Rob L, Halaska M, et al. Dose-dense neoadjuvant chemotherapy followed by sentinel node mapping and laparoscopic pelvic lymphadenectomy and simple trachelectomy in cervical cancer: update results. International Journal of Gynecologic Cancer 2019;29:A12-A13.

‘’They received 3 cycles of NAC in ten-days interval (cisplatin 75 mg/m2, ifosfamide 2g/m2 (max. 3g) in squamous cancers, cisplatin 75 mg/m2, doxorubicin 35 mg/m2 in adeno and adenosquamous cancers).’’

Here dose dense was cisplatin combination (75/m2) every 10 days

C-For the NACCCMA meta-analysis of 2003 [reference 14], that was cited by authors and compared to current results

‘’A substantial amount of heterogeneity was explained by separate analyses of groups of trials. Trials using chemotherapy cycle lengths of 14 days and shorter (Hazard Ratio (HR))=0.83, 95% Confidence Interval (CI)=0.69-1.00, P=0.046) or cisplatin dose intensities greater than or equal to 25 mg/m2 per week (HR=0.91, 95% CI=0.78-1.05, P=0.20) tended to show an advantage for neoadjuvant chemotherapy on survival. Despite some unexplained heterogeneity, the timing and dose intensity of cisplatin-based neoadjuvant chemotherapy appears to have an important impact on whether or not it benefits women with locally advanced cervical cancer and warrants further exploration.

C- Li et al, 2022 (PMID: 35012634), were describing a design for a prospective phase III trial for neoadjuvant chemotherapy in cervical cancer. ‘In the study arm, patients will receive dose-dense cisplatin (40 mg/m2) and paclitaxel (60 mg/m2) weekly for 4 cycles’ they used the term dose dense to describe weekly cisplatin that would bring total cisplatin in 3 weeks to 120mg/m2 which is around double the standard dose (50-70 mg/m2). In the discussion part: ‘’Thus, administering NACT at shorter intervals (dose-dense) may result in enhanced cell death and overcome accelerated repopulation. A dose-dense (weekly) schedule is likely to result in improvement in the outcomes.’’

D- Ferrandina, 2019 PMID: 30506402 ‘This phase II study investigated activity of dose-dense paclitaxel/platinum before radical surgery (RS) in LACC patients…NACT (paclitaxel: 80 mg/m2) and carboplatin (AUC 2) were administered for 6 weeks.’

2] As stressed by authors, since neoadjuvant chemotherapy was not compared against standard of care, the strong results for this study are hypothesis generating that needs validation in prospective trials. The only clinically useful recommendation is the feasibility of giving neoadjuvant chemotherapy before surgery if radiotherapy is not reachable. This recommendation needs to be stated with care, since only 7/22 included studies investigated this option and thus, this recommendation is based mainly on the subgroup analysis performed by authors for these 7 studies.

3] For the endpoint PFS that is now clearly defined by authors in this version. PFS needs to be clearly mentioned in abstract (methods, line 24 and results, line 33). PFS outcomes need to be clearly pronounced in the discussion and compared with PFS/DFS outcomes in the preceding meta-analyses. Besides, as clearly stated in my previous review, at least sub-group analyses are needed for PFS for surgery cases and for dose-dense protocols (cisplatin>105 mg/m2) to further support the conclusion and recommendation. If PFS data is limited I suggest removal of the data for PFS as for locally advanced cervical cancer OS is the most common endpoint used.

4] My recommendation for PFS subgroup analysis would make the discussion more informative because this can be compared to the Ye et al.,2020 (reference 50) who have significant OS and DFS benefit in a subgroup analysis for patients who received neoadj chemotherapy followed by surgery with a median follow up >60 months.

5] I suggest adding the EORTC 55994 design and prelim data [DOI: 10.1200/JCO.2019.37.15_suppl.5503] in the discussion part and I understand that it was not included in the meta-analysis because prelim results are in abstract form and are not included in pubmed. EORTC 55994 is the only modern prospective trial comparing modern neoadjuvant chemotherapy (Cisplatin of at least 225mg/m2, given every 3 weeks) followed by surgery vs standard of care.

Again, I really appreciate the clean methodology and stats used in this well wrought meta-analysis that was further improved in this revised version.
